# Auditory cortical activity elicited by infrared laser irradiation from the outer ear in Mongolian gerbils

Yuta Tamai[1], Yuki Ito[1], Takafumi Furuyama[1,2], Kensuke Horinouchi[1], Nagomi Murashima[1], Itsuki Michimoto[3], Ryuichi Hishida[4], Katsuei Shibuki[4], Shizuko Hiryu[1], Kohta I. Kobayasi[1]*

1 Faculty of Life and Medical Sciences, Doshisha University, Kyotanabe, Kyoto, Japan, 2 Department of Physiology, Kanazawa Medical University, Uchinada, Ishikawa, Japan, 3 Faculty of Science and Engineering, Doshisha University, Kyotanabe, Kyoto, Japan, 4 Department of Neurophysiology, Brain Research Institute, Niigata University, Niigata, Niigata, Japan

* kkobayas@mail.doshisha.ac.jp

**Data Availability Statement:** All relevant data are within the paper and its Supporting Information files.

## Abstract

Infrared neural stimulation has been studied for its potential to replace an electrical stimulation of a cochlear implant. No studies, however, revealed how the technic reliably evoke auditory cortical activities. This research investigated the effects of cochlear laser stimulation from the outer ear on auditory cortex using brain imaging of activity-dependent changes in mitochondrial flavoprotein fluorescence signal. An optic fiber was inserted into the gerbil's ear canal to stimulate the lateral side of the cochlea with an infrared laser. Laser stimulation was found to activate the identified primary auditory cortex. In addition, the temporal profile of the laser-evoked responses was comparable to that of the auditory responses. Our results indicate that infrared laser irradiation from the outer ear has the capacity to evoke, and possibly manipulate, the neural activities of the auditory cortex and may substitute for the present cochlear implants in future.

## Introduction

Cochlear implants allow individuals with hearing impairment to regain auditory sensations by stimulating the cochlear spiral ganglions and bypassing defective hair cells. Such devices have been implanted in more than 300,000 people worldwide (NIH Publication 00–4798). However, they have several limitations, particularly the need for surgical implantation and limited replication of spectral information. Some individuals experience complications following surgery [1] and speech recognition in noisy environments is relatively poor [2]. These problems stem from the limitations of electrical stimulation. Conventional cochlear implants need an electrode array in contact the spiral ganglia and the electrical current spreads widely across the cochlea.

Infrared neural stimulation is a possible alternative to electrical stimulation. Laser neural stimulation can elicit spatially confined neural responses without touching the tissue [3, 4].

**Funding:** This research was supported by Japan Society for the Promotion of Science, grants-in-aid for scientific research KAKENHI Grant number: 17H01769 to KIK; 19K16192 to TF; 18J21644 to YT Funders don't play any role in the study design, data collection and analysis, decision to publish, and preparation of the manuscript.

**Competing interests:** The authors have declared that no competing interests exist.

Several researchers have focused on the physiological mechanism of laser-evoked neural responses. Wells et al. [5] reported that optical absorption of water causes the temperature to rise in cells, and opens heat-sensitive ion channels. The rapid temperature rise reversibly alters the electric capacitance of the plasma membrane, causing cell depolarization [6]. Additionally, the laser-evoked responses of sensory neurons are related to the heat-sensitive TRPV4 channel [7]. The practical application of this technique has been evaluated using various nervous systems, including the sciatic nerve [8], visual cortex [9], and somatosensory cortex [10]. Cochlear implants are one of the most extensively investigated applications of infrared stimulation [11–20]. Izzo and colleagues [21] reported that the auditory nerve can be activated by infrared laser irradiation. Moreno et al. [22] further evaluated the detailed relationship between the irradiation site of the cochlea and the evoked neural response in the inferior colliculus (IC) of guinea pigs. Richter and colleagues demonstrated that the best frequencies of laser evoked responses matched the stimulation site well and that the spatial extent of activation evoked by the laser in the IC was comparable to that produced by tone pips [23].

However, no studies have described the auditory cortical responses evoked by laser stimulation of the peripheral auditory system, although the auditory cortex plays an important role in perception of complex sounds [e.g., 24–26]. Evaluating how laser stimulation affects cortical activities is an essential step towards the ultimate goal of reconstructing the function of the higher auditory systems by using a laser to stimulate the cochlear in patients with hearing impairment.

In this study, we recorded the laser-evoked responses of the cerebral cortex by using flavoprotein fluorescence imaging. This imaging method is based on activity-dependent changes in endogenous fluorescence derived from mitochondrial flavoproteins *in vivo* [27]. This imaging method has been used to detect neural responses to visual [28–30], somatosensory [31–33], and auditory [34–39] stimuli in the primary sensory areas in rodents. Because this technique permits uniform observation of a wide cortical surface, the regional borders between different sensory cortical regions are easily identified by observation of cortical responses elicited by sensory stimulation. Therefore, we examined the auditory cortical responses to laser stimulation of the auditory nerve by comparing laser-evoked cortical fluorescence changes with sensory responses to visual and auditory stimuli.

## Results

### Cochlear laser stimulation evoked auditory cortical activities

The infrared laser was applied to the lateral side of the cochlea through the tympanic membrane (Fig 1A). A clear cochlear microphonic (CM) followed by cochlear responses was evoked by the auditory stimulus (Fig 1B). The mean CM peak amplitude was 99±15 μV (mean ±standard deviation, n = 9). The laser stimulation, on the other hand, did not induce a CM exceeding the noise amplitude (SD of noise during prestimulus period: 7.10 μV). The brain areas activated by visual, auditory or the laser stimulation are shown in Fig 2. Visual stimuli mainly activated the occipital region, whereas auditory (noise burst) and laser stimuli (7.4 mJ/cm$^2$) activated the temporal regions (Fig 2A and 2B). To quantify the amplitudes of each response, region of interests (ROIs) were placed at the center of the area responding to the visual (ROI I) and auditory (ROI II) stimuli. Visual stimuli evoked higher mean response amplitudes at ROI I than at ROI II (ROI I vs ROI II: 0.54±0.07% vs 0.19±0.04%, Mean±SD), whereas auditory stimuli evoked higher mean response amplitudes at ROI II than at ROI I (ROI I vs ROI II: 0.21±0.04% vs 0.62±0.08%). With laser stimulation, the mean response amplitudes at ROI I and ROI II were 0.16±0.04% and 0.32±0.06%, respectively. The difference between ROI I and ROI II was statistically significant for each stimulus (all $P < 0.001$; Fig 2C).

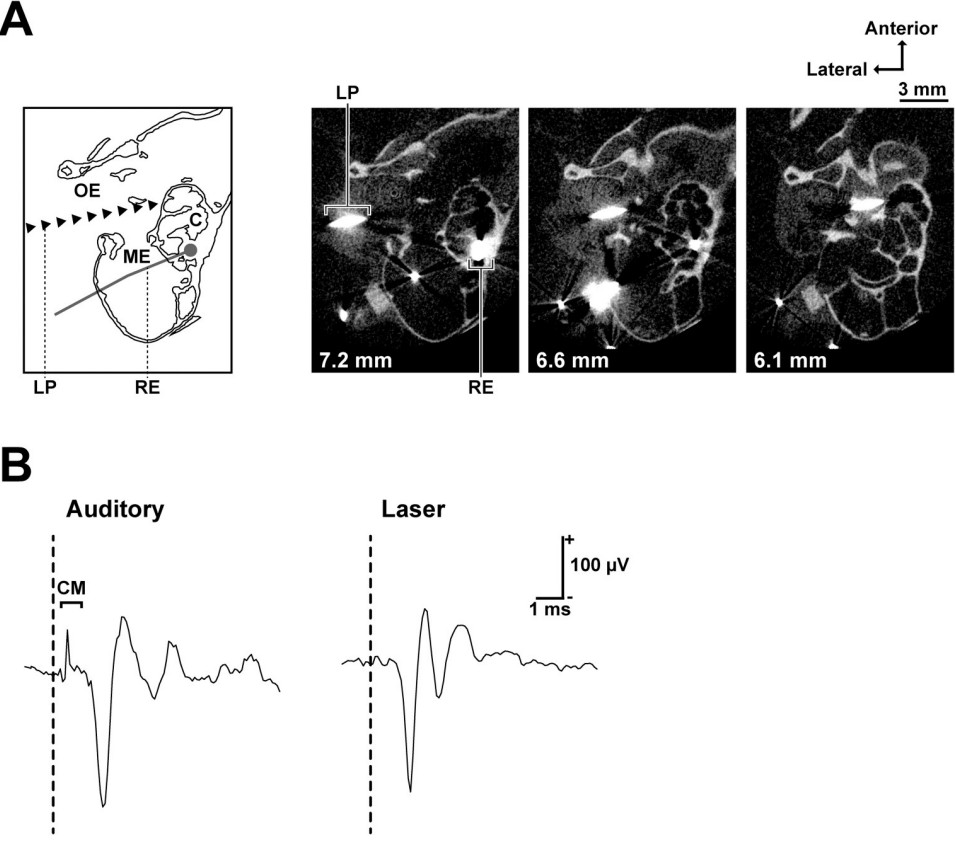

**Fig 1. Laser irradiation site and the cochlear responses to laser and auditory stimuli.** (A) Schematic of recording and stimulation configuration (the left-most) and microcomputed tomography images of a stimulated cochlea (horizontal section). The distance from the sagittal suture is stated on the bottom left of each image. The recording electrode (silver wire) and the laser path (tungsten wire) are shown as larger than their actual sizes due to metal artifacts in CT images. (B) Cochlear responses to auditory (100 µs single click with 70 dB peak-to-peak equivalent SPL) and laser (100 µs single pulses, 4.9 mJ/cm²) stimuli. A clear cochlear microphonic (CM) was observed only after the auditory stimulus. The vertical dashed lines show stimulus onset. Abbreviations: OE, outer ear; ME, middle ear; C, cochlea; LP, laser path; RE, recording electrode; CM, cochlear microphonic.

The summarized ($n$ = 11) temporal areas activated by the laser are shown in Fig 2D. In each individual, the areas with responses >75% of the maximum amplitude were defined as activated regions and the individual activated regions were superimposed on each other to produce the summarized images (Fig 2D). To obtain the entire outline of the auditory cortex, the regions that responded to various auditory stimuli (1, 4, 20, and 50 kHz tone bursts and 1–20 kHz band noise) were overlaid. The summarized auditory- and laser-evoked brain areas were 12.6 and 7.7 mm², respectively. The auditory-evoked brain area contained the entire laser-evoked brain area.

## The laser-evoked cortical responses were comparable to the auditory-evoked responses

Fig 3 shows the time-course of the auditory- and laser-evoked responses of the temporal region and the intensity-dependent changes. After presenting the auditory stimulus (85 dB SPL, 4 kHz click train), responses were observed in broad temporal regions. The response peaked at around 1.0 s, and then gradually decreased to near the prestimulus baseline (Fig 3A upper). A similar temporal profile was observed for the laser stimuli. Laser stimulation at 7.4 mJ/cm², 4

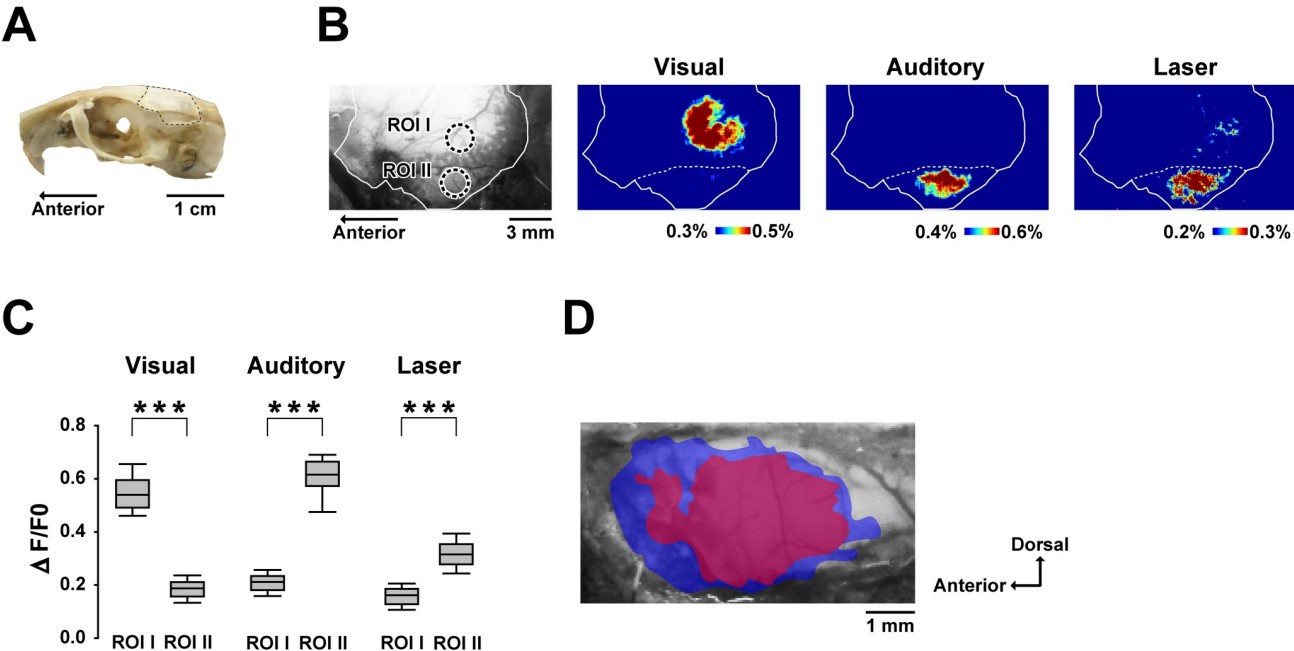

**Fig 2. Cortical regions activated by visual, auditory and laser stimuli.** (A) Photograph showing the recording site (dashed line). (B) An original fluorescence image showing the ROI and pseudocolor images of $\Delta F/F_0$ of the left hemisphere. The solid line indicates the recording site and the dashed line indicates the ridge of the parietal bone. Each pseudocolor image is averaged over a period of 500 ms around the peak response time. (C) Amplitudes of visual-, auditory-, and laser-evoked responses for each ROI. The positions of ROI I and ROI II, depicted in (B), are centered over the peak responses corresponding to the visual and auditory stimuli, respectively. The box plots show the median and 25th and 75th percentiles, with whiskers at the 10th and 90th percentiles. ***$P < 0.001$ (Mann–Whitney $U$ test with Bonferroni correction for multiple comparisons). (D) Positional relationship between the auditory- (blue) and laser-evoked (red) areas of the temporal region. The colored areas superimposed on the original fluorescence image show the cortical regions with a 75% maximum $\Delta F/F_0$ response. Data from 11 subjects were aligned based on the cerebral vascular pattern and were pooled.

kHz pulse train, evoked responses from relatively broad temporal regions, peaking at around 1.0 s after stimulus onset, and then gradually decreased to the baseline (Fig 3A lower). Increasing the stimulus intensity of both auditory and laser stimulation produced systematic change in the response amplitude (Fig 3B). Evaluation of the time-courses of the responses of ROIs revealed a steady increase in peak amplitude and a decrease in latency with increasing intensities of the auditory or laser stimuli (Fig 3C). The amplitude and latencies of the maximum responses, defined as the time to reach 50% maximum of the least observable response, were measured and are shown in Fig 3D ($n = 6$). As indicated, the amplitude increased significantly ($r = 0.54$, $P < 0.05$), and the latency decreased significantly ($r = -0.49$, $P < 0.05$) as the intensity increased (Fig 3D). The mean latencies to reach the 25%, 50%, 75%, and 100% maximum responses elicited by the 7.4 mJ/cm² laser were compared with the auditory responses, which exhibited equivalent peak responses in each gerbil (average stimulus intensity: 81.4 ± 2.3 dB SPL, $n = 12$). The mean latencies to the 25%, 50%, 75%, and 100% maximum responses were similar for the laser (0.17, 0.35, 0.49 and 0.75 s) and auditory (0.16, 0.29, 0.43 and 0.77 s) stimuli (Fig 3E).

## Discussion

Our results indicate that laser irradiation of the cochlea elicited flavoprotein fluorescence responses in restricted temporal regions (Fig 2). Flavoprotein fluorescence imaging has been used for precise mapping of sensory cortical areas including the visual and auditory cortices [37–44]. We examined whether laser stimulation of the cochlea evoked auditory cortical

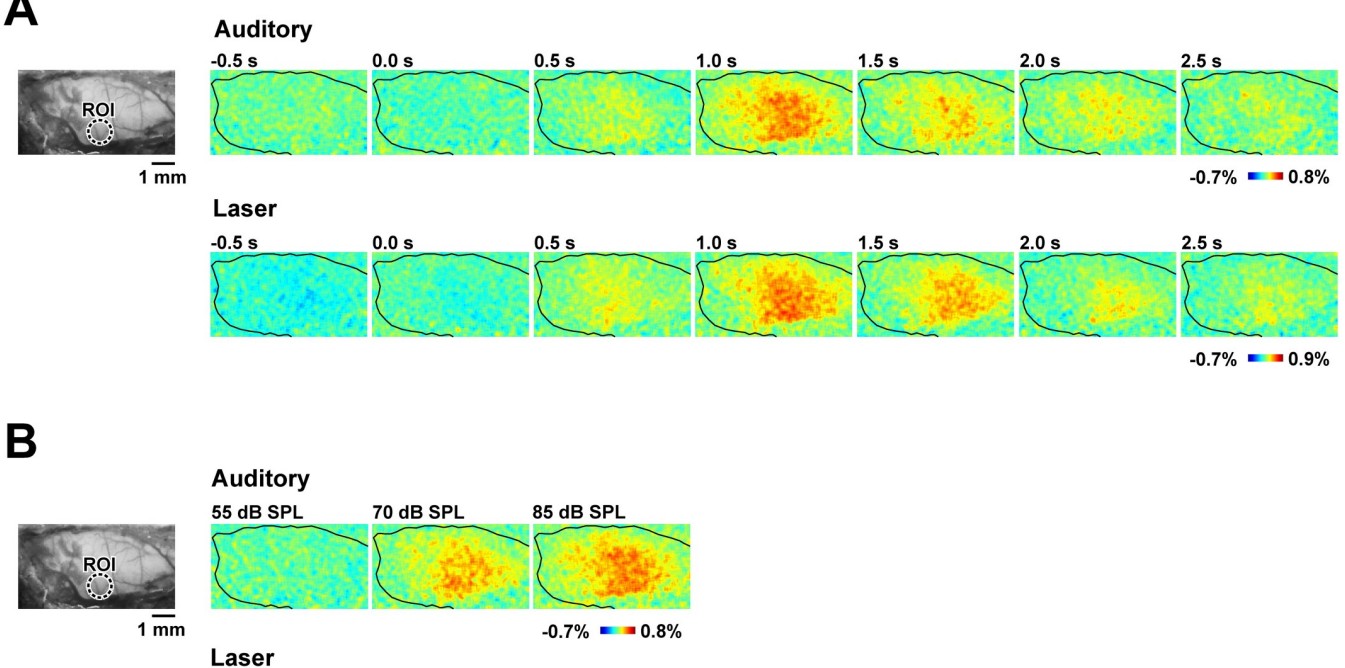

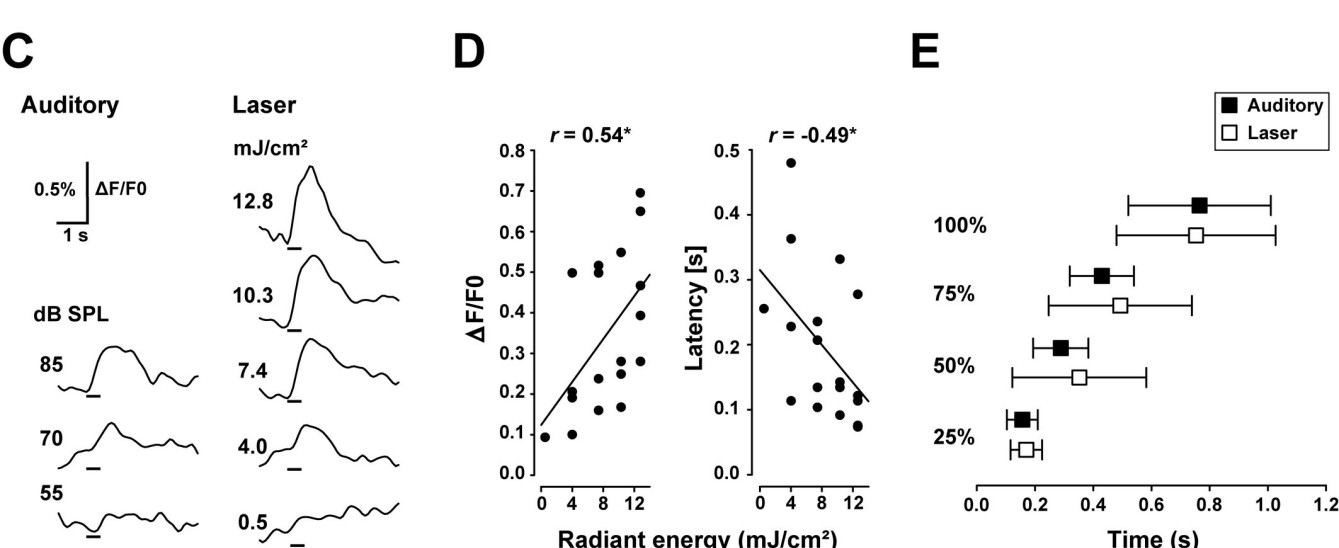

**Fig 3. Auditory- and laser-evoked flavoprotein responses in the temporal cortex.** (A) Original fluorescence image (left) and flavoprotein responses evoked by auditory (85 dB SPL) and laser (7.4 mJ/cm²) stimuli (right). The circular window shows the region of interest (ROI) centered on the maximum responses of all recordings. The recording areas are indicated with black lines. (B) Original fluorescence image and changes in intensity-dependent responses to auditory (55–85 dB SPL) and laser (0.5–12.8 mJ/cm²) stimuli. (C) Time-course of the fluorescence changes in the ROI following acoustic (55–85 dB SPL) and laser (0.5–12.8 mJ/cm²) stimuli. The vertical bars on the bottom represent the stimulus period. (D) Changes in response amplitude and response latency following infrared laser stimulation of various radiant energy (*n* = 6). (E) Latencies to the 25%, 50%, 75% and 100% maximum response to laser (7.4 mJ/cm²) and equivalent auditory stimuli (*n* = 12). Error bars indicate the standard deviation.

responses by comparing the responses to visual and auditory stimuli. Different brain areas were activated by the visual and auditory stimuli (Fig 2B), confirming the location of each sensory region. The responses of ROIs located in the visual (ROI I) and auditory (ROI II) cortices exhibited significant modal specificity (Fig 2C), and laser stimulation evoked significantly greater responses at ROI II than at ROI I, suggesting that cochlear laser irradiation activated the auditory cortical areas.

The outline of the auditory cortical regions activated by a wide range of sound stimuli extended about 4 mm anterocaudally and 3 mm ventrodorsally (Fig 2D). This spread corresponds to the auditory cortices described in several electrophysiological studies, including ours [45–47]. Fig 2D shows that the laser-evoked brain area was enclosed by the auditory-evoked area, which indicates that non-auditory regions were not stimulated by laser irradiation of the cochlea. In addition, the laser-evoked area showed a wide distribution within the auditory-evoked area (Fig 2B and 2D), which implies that the laser stimulus activated a wide frequency region of the primary auditory cortex. CT images of the irradiation site (Fig 1A) provide supporting evidence that the laser was centered on the second turn of the cochlea and the beam covered neighboring turns (apical and basal region). Therefore, the laser stimulated a relatively wide frequency region of the auditory periphery. In future studies, we envisage using a more focused beam with precise position control to help to manipulate the activities of more specific auditory cortical regions (i.e., different frequency regions), as previously tested in the inferior colliculus [22, 23].

Generally, a stronger auditory stimulus elicits a neural response with a shorter latency and greater magnitude [e.g., 48, 49]. Takahashi et al. reported that a stronger 5 kHz tone burst elicited a stronger flavoprotein response in the range of 40 to 80 dB SPL [37]. We obtained similar patterns with laser stimulation (Fig 3C and 3D). The temporal profile of the laser-evoked flavoprotein response was also similar to that of the auditory-evoked response (Fig 3A and 3E). The latencies for each amplitude (25%, 50%, 75%, and 100% maximum peak) were not statistically significant between the modalities (Mann–Whitney $U$ test, All $P$ >0.1). These results suggest that laser-evoked cortical activity is similar to auditory-evoked cortical spiking activity in terms of the temporal structure and intensity dependence. It is, however, noted that the temporal characteristics of the cortical spiking activity under urethane anesthesia generally differs from that of awake animals in several modalities (visual [50], auditory [51, 52] and somatosensory [53]). A similar result was found by Yanagawa and colleagues [34] who used a flavoprotein fluorescence recording. They found that the auditory evoked cortical fluorescence signal had a longer response latency and lasted longer under urethane anesthesia compared to that under ketamine anesthesia. Therefore, when interpreting the results of the current study, differences of temporal patterns in the cortical response between the reported ones and that of awake animals should be taken into consideration.

While our data demonstrated that cochlear laser stimulation evoked auditory cortical responses, many challenges regarding its feasibility in deaf patient remain. Some research groups reported that the auditory response induced by the cochlear laser stimulation significantly decreased after chemical deafening (i.e. neomycin administration) which deteriorate hair cells [16, 18, 54]. Their results indicate that the laser needs functional hair cells to stimulate cochlear spiral ganglion neurons; therefore, the method might be applicable only to some types of conductive hearing loss but not to sensorineural hearing loss. Tan et al. [20], on the other hand, revealed the laser stimulation could elicit auditory neural activity in congenitally deaf mice, whose cochlea has no synaptic transmission between inner hair cells and spiral ganglion neurons, suggesting that the laser can potentially work as an alternative to the electrical cochlear implant. The idea is partially supported by our data (Fig 1B): the laser stimulation is capable of evoking CAP without causing a measurable CM response. Future studies involving

subjects with several causes of hearing loss are needed to evaluate the efficacy of the laser stimulation method.

This study investigated the cortical responses to laser irradiation of the cochlea. Flavoprotein imaging revealed that the laser stimuli activated the auditory cortex, and that the laser-evoked responses showed similar temporal profiles and intensity dependence to auditory-evoked responses. These results indicate that laser-evoked neural signals could be processed in the auditory cortex and may elicit auditory perception. If multiple fibers can be inserted into the ear canal to stimulate tonotopically separated areas of the auditory cortex, the use of infrared laser stimulation could be a good alternative to the present cochlear implants. Because infrared laser irradiation does not require surgery, it will be much easier for patients with hearing loss to try this method before surgical operation for cochlear implants.

## Materials and methods

### Animals

All experimental procedures were performed in accordance with guidelines established by the Ethics Review Committee of Doshisha University, and the experimental protocols were approved by the Animal Experimental Committee of Doshisha University. Twenty-four experimentally naive Mongolian gerbils (*Meriones unguiculatus*), aged 4–12 months, were used in this study. All of the gerbils were bred and reared in our laboratory. Each animal was housed with 2–5 other gerbils in a cage 20 (W) × 40 (L) × 17 cm (H), with free access to food and water. The animal room had a 12-h light–dark schedule, a temperature of 22–23˚C, and relative humidity of approximately 50%.

### Animal surgery

The gerbils were deeply anesthetized by an intraperitoneal injection of urethane (1.5 g/kg). The bulla was exposed by an incision of the muscle and skin from the shoulder to the jaw, and a hole was made on the bulla. A silver electrode (Nilaco, Tokyo, Japan; diameter: 0.13 mm; impedance <11 kΩ) was inserted into the hole and hooked onto the bony rim of the round window of the cochlea (Fig 1A). A reference electrode was placed on the shoulder skin, wet with saline. The pinna was removed to provide a clear sight of the tympanic membrane. The parietal and left temporal area of skin was excised and the temporal muscle over the left auditory cortex was removed after applying a local anesthetic (xylocaine gel, Aspen Japan, Tokyo, Japan). A metal plate was attached using dental cement (Shofu, Kyoto, Japan) to the skull above the right hemisphere, about 1–5 mm posterior from the bregma, and the plate was screwed onto a recording stage to stabilize the gerbil. The surface of the skull was covered with liquid paraffin (Wako, Osaka, Japan) to keep the skull transparent.

### Fluorescence imaging

Fluorescence imaging was performed under urethane anesthesia. A cooled charge-coupled device camera system (BU-61, Bitran, Saitama, Japan) was mounted on a stereoscopic microscope (SZX16, Olympus, Tokyo, Japan) with a 1× objective lens (Olympus, Tokyo, Japan; numerical aperture: 0.15). The gerbil's cortical surface over the visual, somatosensory, and auditory cortex of the left hemisphere was illuminated with blue light ($\lambda = 420$–$490$ nm) using a 130 W mercury lamp for light source (SHI-130 OL, Ushio, Tokyo, Japan) through fiber optic cables. The radiant exposure of blue light was set at 5.7 W/cm$^2$ using a digital power meter (PM100D, Thorlabs, Newton, NJ, US) with a thermal power sensor (S302C, Thorlabs). Cortical images (240 × 135 pixels after binning) of endogenous green fluorescence ($\lambda > 510$ nm)

were recorded for 8 s at 100 ms intervals. The images were averaged over 18–20 trials. Spatial filtering (averaging over $5 \times 5$ pixels) was performed to improve image quality. The baseline intensity ($F_0$) of each pixel was determined by averaging the prestimulus images obtained for 2 s. The fluorescence change ($\Delta F$) was determined as the ratio of the fluorescence signal ($F$) over baseline intensity ($F_0$) for each pixel. The relative fluorescence change ($\Delta F/F_0$) was used to visualize the response in a pseudocolor scale.

## Auditory and visual stimuli

Visual and auditory stimuli were presented to the anesthetized gerbil to locate the primary visual and auditory cortices. Click trains with a repetition rate of 4 kHz, tone burst (frequency: 1, 4, 20, 32, 50 kHz; rise and fall time: 50 ms), and noise burst (frequency: 1–50 kHz) were used for auditory stimuli. The 4 kHz repetition rate was selected as the geological average of their most sensitive frequency range: 1–16 kHz [55]. The click train consisted of a 100 μs rectangular pulse of the same polarity. The auditory stimuli were applied for 500 ms via a loud speaker (FT28D, Fostex, Tokyo, Japan). This trial was repeated 18–20 times at intervals of 25 s. These experimental parameters, including urethane anesthesia, stimulus duration, inter-stimulus interval, and number of repetitions, were used in our previous studies [37–39]. The speaker was placed in front of the gerbil's nostril, and approximately 10 cm apart from their ears. The auditory stimuli were generated using a digital-to-analog converter (Octacapture, Roland, Shizuoka, Japan) with a 192 kHz sampling rate. The sound pressure level varied from 50 to 85 dB SPL (RMS) and was calibrated using a microphone (Type 1, ACO, Tokyo, Japan) placed by the gerbil's head.

A red light-emitting diode (LED) ($\lambda = 620$–625 nm; diameter, 5 mm) was used for the visual stimulus. The LED was placed 3 cm from the gerbil's right eye in the horizontal plane. The light stimulus was applied for 500 ms at an interval of 25 s using a microcontroller (Arduino Uno, Ivrea, Italy).

## Laser stimuli

A repetitive pulsed infrared laser was used to apply the infrared laser stimuli. The individual pulse of laser was rectangular in intensity. Each stimulus pulse was applied for 100 μs and repeated 2000 times at intervals of 0.25 ms to create a 500 ms laser train. The laser stimulation parameters were determined in accordance with parameters that were used in auditory experiments. The laser trains were presented 18–20 times at intervals of 25 s via an optic fiber (diameter = 400 μm; NA = 0.22). The position of the optic fiber was controlled with a micromanipulator (MM-3, Narishige, Tokyo, Japan). The laser stimulation was performed as described in previous studies including ours [21, 56]. The head of the animal was oriented so that the sagittal suture is aligned with the horizontal plane. The optic fiber was inserted mediolaterally into the outer ear canal of the left ear with the tip angled rostrally by 10 degree and dorsally by 5 degree (Fig 1A). The fiber tip was directing at ca. 0.8 mm beneath the posterior malleolar fold, and was placed 1.0 mm short of touching the tympanic membrane. The laser was applied to the lateral side of the cochlea through the tympanic membrane, and evoked responses were monitored with the cochlear electrode (Fig 1B). The placement of the fiber was adjusted dorsoventrally and rostrocaudally to obtain maximum signal intensity. This procedure allowed us to stimulate the second turn of the cochlea with the standard error less than ±350 μm. The pulsed infrared laser was created with a diode laser stimulation system (BWF-OEM, B&W TEK, Newark, DE, US). The wavelength of the infrared laser was 1871 nm. The width (full width at 50% maximum) of the laser beam was 2.2 mm at the radiation site. The radiant energy for each pulse was measured using a digital power meter (PM100D,

Thorlabs, Newton, NJ, US) with a thermal power sensor (S302C; Thorlabs, Newton, NJ, US), and was calibrated to between 0.5 and 12.8 mJ/cm$^2$. The sound pressure level of the sound that is generated by the optoacoustic effect reached 19 dB SPL when measured at maximum intensity (i.e., 12.8 mJ/cm$^2$) at 2 mm from the laser fiber tip. Our previous research assured that continuous laser irradiation with that maximum intensity for 1 h did not cause any thermal damage on the gerbil's cochlea and tympanic membrane [56]. The voltage commands for the laser stimuli were generated using a digital-to-analog converter (Octacapture, Roland, Shizuoka, Japan).

## Microcomputed tomography imaging

Anatomical microcomputed tomography (microCT) images of the stimulation site were obtained using an X-ray microCT system (SMX-160CTS, Shimadzu, Kyoto, Japan). After the recording, the gerbils were overdosed with an intraperitoneal injection of pentobarbital (200 mg/kg). A tungsten wire (25 μm diameter), which was attached under and in parallel with the stimulation glass fiber, was inserted into the middle ear through the tympanic membrane. The wire was cemented to the ear canal with a cyanoacrylate glue and dental cement. The wire followed the optical axis of the stimulus laser (Fig 1A). After postmortem perfusion with phosphate buffer and 4% paraformaldehyde, the formalin-fixed gerbils were decapitated and the cochleae were imaged in air using the X-ray microCT system set at 80 kV and 140 μA. Two-dimensional images (512 × 512 pixels; pixel size: 39 μm) were obtained at 39μm intervals and converted into TIFF format files using TRI/3D-BON (RATOC, Osaka, Japan).

## Data analysis and statistics

The amplitude of the flavoprotein fluorescence response was evaluated as the median of $\Delta F/F_0$ in the region of interest (ROI; circular window with a diameter of 1 or 2 mm), which was centered on the highest response peak of the brain image acquired at 1 s after stimulus onset. The time-course of the fluorescence change was calculated by averaging the response in the ROI. The response amplitude was defined as the difference between the maximum and minimum amplitudes between 0 and 1.5 s after stimulus onset. Response latency was evaluated as the time at which the signal reached a threshold, which was defined in each gerbil as a half of the peak amplitude of the smallest response. The Mann–Whitney $U$ test was used to evaluate statistical significance and Bonferroni correction was applied when necessary. Correlation coefficients were obtained using Pearson's correlation analysis. All analyses were performed using MATLAB (MathWorks, Natick, MA, USA) and RStudio (Rstudio Team, 2018).

## Supporting information

**S1 File.**
(ZIP)

## Acknowledgments

We wish to thank Hiroshi Riquimaroux, Suguru Matsui, Hidetaka Yashiro, Shoko Nakanishi, Koichi Sawada, Mami Matsukawa, Kohei Yoshitake, Hiroaki Tsukano and Olga Heim for their technical support and valuable advice regarding experimental design.

## Author Contributions

**Conceptualization:** Yuta Tamai, Takafumi Furuyama, Kohta I. Kobayasi.

**Data curation:** Yuta Tamai, Yuki Ito, Takafumi Furuyama, Kensuke Horinouchi, Nagomi Murashima, Itsuki Michimoto.

**Formal analysis:** Yuta Tamai, Yuki Ito.

**Funding acquisition:** Kohta I. Kobayasi.

**Investigation:** Yuta Tamai, Yuki Ito.

**Methodology:** Yuki Ito, Takafumi Furuyama, Ryuichi Hishida, Katsuei Shibuki.

**Project administration:** Kohta I. Kobayasi.

**Software:** Yuta Tamai.

**Supervision:** Shizuko Hiryu.

**Validation:** Yuta Tamai, Yuki Ito.

**Visualization:** Yuta Tamai.

**Writing – original draft:** Yuta Tamai, Kohta I. Kobayasi.

**Writing – review & editing:** Yuta Tamai, Ryuichi Hishida, Katsuei Shibuki, Shizuko Hiryu, Kohta I. Kobayasi.

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
