## [Decision Letter · Decision Letter 0]

10 Aug 2020

PONE-D-20-17518

Auditory cortical activity elicited by infrared laser irradiation from the outer ear in Mongolian gerbils

PLOS ONE

Dear Dr. Kobayasi,

Thank you for submitting your manuscript to PLOS ONE. After careful consideration, we feel that it has merit but does not fully meet PLOS ONE’s publication criteria as it currently stands. Therefore, we invite you to submit a revised version of the manuscript that addresses the points raised during the review process.

We look forward to receiving your revised manuscript.

Kind regards,

Michael Smotherman

Academic Editor

PLOS ONE

Additional Editor Comments:

Dear Kohta,

I apologize for the delay in getting this back to you. It has been challenging to get prompt responses from reviewers this summer, presumably due to the ongoing pandemic. In this case both reviewers were generally positive about your manuscript and only offered minor suggestions for improvements. I do not think the paper needs more data, as hinted at by reviewer #2, so just focus on the minor stuff and the manuscript should be good to go.

Sincerely,

Mike

Journal Requirements:

Reviewers' comments:

Reviewer's Responses to Questions

**Comments to the Author**

1. Is the manuscript technically sound, and do the data support the conclusions?

Reviewer #1: Yes

Reviewer #2: Yes

2. Has the statistical analysis been performed appropriately and rigorously? 

Reviewer #1: No

Reviewer #2: Yes

3. Have the authors made all data underlying the findings in their manuscript fully available?

Reviewer #1: Yes

Reviewer #2: Yes

4. Is the manuscript presented in an intelligible fashion and written in standard English?

Reviewer #1: Yes

Reviewer #2: Yes

5. Review Comments to the Author

Reviewer #1: I have read the manuscript entitled “Auditory cortical activity elicited by infrared irradiation from the outer ear in Mongolian gerbils” by Tamai et al. The paper explores the effects of cochlear laser stimulation on the auditory cortex and overall, it is well-written and easy to read. In summary, the authors used brain imaging of activity-dependent changes in mitochondrial flavoprotein fluorescence signal to show that cochlear laser stimulation activates the primary auditory cortex. In addition, they showed that cortical activity evoked by laser stimulation is similar to that evoked by acoustic stimulation. I believe this paper to be highly relevant for PLoS One. The paper is, overall, written concisely and it needs only a few corrections or improvements, which are listed below.

Introduction:

Page 4. Line 60. It would be helpful to the reader if the author summarized the results by Moreno et al. 2011 and Richter et al. 2011 in the inferior colliculus.

Results:

Page 6. Line 83. The data presented in Figure 1is not well described in the text. Please, provide a better description of Fig. 1. For example, it is hard for a general reader to see the cochlear microphonic observed only with the auditory stimuli. The authors should provide a statistical comparison between the cochlear responses to laser and auditory stimuli.

Page 6. Line 86. Define region of interest (ROI) at the first use.

Discussion:

Page 29. Line 141-147. Move to Results section.

The authors should briefly discuss the effects of anesthesia on the temporal characteristics of the cortical activity.

Reviewer #2: In this manuscript, the authors show that stimulation of the cochlea with an infrared laser evoked widespread neural activation of auditory cortex. This is to be expected, since as the author’s point out several papers have previously described the use of a laser to excite the auditory nerve. But it is also true that this is the first study to measure the neural response at the level of the cortex, which may become important if this technology emerges as a central component of future hearing aids. At present, it seems that the laser stimulates too broad an area of cochlea to be useful in this regard, but there are other issues to explore and this evolving technique appears to be gaining interest. My main criticism of the paper is that is doesn’t seem to be testing anything more that whether or not laser activation of the cochlea leads to activation of the cortex. Instead, the focus seems to be optimizing the laser stimulus parameters for replicating the responses to sounds. The paper is technically sound though, and this data may be useful for future studies even though it isn’t especially interesting on its own.

That the laser apparently uses heat to stimulate neurons evokes questions about how stimulation intensity, rate and duration could impact long-term efficacy. The authors cite [51] evidence that 1-hour continuous stimulation did not cause thermal damage, but this manuscript is lacking information about whether the laser evoked responses follow patterns of adaptation or other temporal dynamics similar to acoustic responses. It would have been preferable to see some investigation of how repeated or prolonged stimulus regimes influenced subsequent responses? Did the excitation adapt or decay over time? Or, if the goal was to see how well the laser can replicate natural acoustic responses, surely there could have been more to explore than simply asking how laser power impacts latency.

Minor

Line 24; it’s should be its

Line 106 and 232: the 4kHz click train seems like an unusually fast acoustic stimulus presentation rate. Why were such fast rates used?

Likewise, what was the rationale for this particular laser stimulus pattern (100 us, 4 kHz, 500 ms, every 25 s x 20 repetitions). Since this paper does not describe how these parameters influence response properties, perhaps adding a more details about the rationale could clarify this.

Line 165-166 indicates that the laser needs hairs cells to stimulate spiral ganglion neurons, but the fig 1 legend suggests hair cell receptor potentials are not contributing to the laser evoked response. Which is it? Can the authors elaborate on the significance of the absent CMP?

6. PLOS authors have the option to publish the peer review history of their article (what does this mean?). If published, this will include your full peer review and any attached files.

Reviewer #1: No

Reviewer #2: No

---

## [Author Response · Author response to Decision Letter 0]

15 Sep 2020

For our response, please see the 2nd cover letter file: "Response to Reviewers (LetterToEditor200912-2.docx)".

---

## [Editor Report · Decision Letter 1]

23 Sep 2020

Auditory cortical activity elicited by infrared laser irradiation from the outer ear in Mongolian gerbils

PONE-D-20-17518R1

Dear Dr. Kobayasi,

We’re pleased to inform you that your manuscript has been judged scientifically suitable for publication and will be formally accepted for publication once it meets all outstanding technical requirements.

Kind regards,

Michael Smotherman

Academic Editor

PLOS ONE
---

## [Editor Report · Acceptance letter]

29 Sep 2020

PONE-D-20-17518R1 

Auditory cortical activity elicited by infrared laser irradiation from the outer ear in Mongolian gerbils 

Dear Dr. Kobayasi:

I'm pleased to inform you that your manuscript has been deemed suitable for publication in PLOS ONE. Congratulations! Your manuscript is now with our production department. 

Kind regards, 

on behalf of

Dr. Michael Smotherman 

Academic Editor

PLOS ONE